# Assessment of Genetic and Health Management of Tunisian Holstein Dairy Herds with a Focus on Longevity

**DOI:** 10.3390/genes14030670

**Published:** 2023-03-08

**Authors:** Chaima Sdiri, Ikram Ben Souf, Imen Ben Salem, Naceur M’Hamdi, Mohamed Ben Hamouda

**Affiliations:** 1Research Laboratory of Ecosystems & Aquatic Resources, National Agronomic Institute of Tunisia, Carthage University, 43 Avenue Charles Nicolle, Tunis 1082, Tunisia; 2Department of Animal Production, Ecole Nationale de Médecine Vétérinaire de Sidi Thabet, Sidi Thabet 2020, Tunisia; 3Institut National de la Recherche Agronomique (INRAT), Rue Hédi Karray, El Menzah 1004, Tunisia

**Keywords:** longevity, health, genetic, genomic selection, Tunisian Holstein

## Abstract

In Tunisia, the recognition of the possibility of including longevity and disease resistance in dairy cattle selection objectives has been hypothesized as a useful strategy by both researchers and producers. However, in this paper, the state of the art, with a focus on health and longevity, is reviewed. Along the same lines, the heritability for the milk traits, fertility traits, and longevity of Tunisian Holstein dairy cows complies with the literature. Therefore, the influence of genetics on some diseases of the dairy cow was investigated. In addition, a decreasing efficiency in cow fertility has been observed over the last few years. The results showed that the risk of culling increased with common diseases. When analyzed with the Weibull model, functional lifespan was strongly influenced by milk yield; therefore, the risk increased with a reduced milk yield. In her first three lactations, the relative risk of selection increased gradually with lactation. Thus, the risk of thinning is highest at the beginning and end of the first feeding and the end of her second feeding. In conclusion, the risk of culling was reduced in parity. The factors that influence the life of the herd, such as health, husbandry, environmental conditions, and management, are often ignored when evaluating longevity.

## 1. Introduction

For the modern dairy cow, advances in genetics and breeding for productivity have resulted in an increased incidence of health disorders and reduced longevity. However, to maintain farm sustainability, farmers need to optimize the balance between maximum production and minimum production costs [1]. Moreover, reduced profitability is associated with the costs of dairy herd health and fertility, which are also major causes of involuntary culling. Nevertheless, reducing the incidence of disease in dairy cows is of economic, social, and environmental importance. Therefore, dairy cattle selection around the world has focused on increasing milk production due to consumer demand and the impact of production on farm profit margins. This was extremely successful through the combination of genetic selection with improvements in nutrition and health management. On the other hand, Oltenacu and Broom [2] indicated that inefficiencies exist because the increased production has led to negative effects on health, reproduction, and longevity. Miglior et al. [3] insisted on the inclusion of functional traits, such as fertility, health, and longevity, and reported that these traits have both economic and socioeconomic impacts through improving animal welfare and the sustainability of dairy production.

Brickell and Wathes [4] suggested that extending the productive life of cows reduces replacement costs, allows for more limited selection, and increases the potential milk yields from adult cows, thus improving milk production. Generally, the culling rate was higher for low-producing cows and older ages at first calving. In addition, higher-producing cows are found to be culled at an earlier age than low-producing ones. M’Hamdi et al. [5] concluded that cows at the start and end of the first lactation and at the end of all other lactations were at the highest risk of culling. Tunisian dairy herds reported an average culling rate of 15–22% [5], with more than 50% of the cullings resulting from involuntary causes such as infertility, mastitis, and lameness [5,6]. In addition, Heikkilä et al. [7] reported that mastitis and lameness are the costliest diseases for dairy farmers as they cause sharp declines in milk production and farm income. Agiri et al. [8] reported that the true herd lifespan of Tunisian Holstein-Friesian cattle averaged 41.99 months, corresponding to a production life of 3.5 years, and the number of cows culled after the first two lactations reported that only 7.14% of cattle remained, compared to 57% in their fifth lactation. The genetic improvement program of Tunisian Holstein dairy cow has been elaborated since the 1960s to strengthen this sector; recording, performances control, AI, and Herd-Book are implemented and performed by the «Office de l’Elevage et des Pâturages (OEP)» [9]. Large phenotypic databases are continuously evaluated in collaboration with universities to track the pedigree of herds of animals, register their performance, and use them to implement herd genetic tendencies and management. This paper reviews the main results of the genetic and health management of Tunisian Holstein cattle regarding the situation across the world.

## 2. Animal Health, Diseases, and Welfare

The health and welfare of all animals in the dairy herd are paramount for profitable and efficient milk production. The disease is often measured by economic impact, but animal health is also part of animal welfare. The pain and discomfort caused by health problems affect an animal’s well-being, so animals must be in good health to improve animal welfare [6]. Records of the incidence and prevalence of various diseases are made more readily available through farm record-keeping systems. Producers must be able to correctly identify specific animal health problems early to improve animal welfare and herd health. Unfortunately, in Tunisia, there is no systematic recording system of health events in dairy herds [5]. This lack of information concerning health problems and their impact on dairy cow productivity prohibits work on the genetic analysis of the health and diseases of the dairy herd [5]. However, evidence of culling can partially compensate for this lack of information. Examining data for screening can be a cost-effective alternative compared to the costs of collecting, storing, and analyzing data on health disorders. Mastitis and lameness are major problems in Tunisian herds. Mastitis is a production, food quality, and safety issue. From an animal welfare perspective, it is a localized and painful infection for cows that can cause systemic illness leading to fever, dehydration, depression, and even death, depending on the type of infection and the cow’s resistance. However, mastitis has been assessed during somatic cell counting. A prospective study of 21 selected dairy farms in northern Tunisia was conducted by Mtaallah et al. [10] to assess the reduction in milk yield due to high somatic cell counts in bulk tanks and to find associations between the risk factors and asymptomatic mastitis. The authors found that the average somatic cell count of bulk milk was 626,103 cells/mL and the average milk loss due to the somatic cell count in bulk tanks was 524 kg per cow per year. The risk factors associated with high in-tank somatic cell counts include: (i) livestock risk factors (inadequate bedding area, inadequate cleaning of bedding and waste areas); and (ii) milking risk factors (tap washing with showers without adjustable flow and individual towels. No wiping, more than five milking shifts per cow-herd; no overspray before milking or milking healthy and mastitis cows at the same time; no teat dipping). M’Hamdi et al. [11] analyzed the data, consisting of 73,189 test-day records of somatic cell counts, for the three first lactations of 8350 Holstein cows calving between 1997 and 2003 in 114 dairy herds. The results showed that the milk yield (MY) was largest in the second control, at 25 ± 9 kg, and lowest in the tenth control (14 ± 6 kg).

Lameness among dairy cows is widely recognized as one of the most serious (and costly) animal welfare issues affecting dairy cattle. Lameness is the third most common infection in cows in modern barns. Environmental factors (diet, stables, injuries) and genetic factors are responsible for this condition. Lameness is also recognized as an important welfare problem, causing pain, and impairing the cow’s ability to exhibit normal behavior [12,13]. In a study conducted by Ferchichi et al. [14] in Tunisian dairy herds, the incidence of lame cows averaged 67%. However, Bouraoui et al. [15] showed that podal pathologies have an incidence equal to 38.71% (score > 2). They reported that the incidence of lameness was approximately 37% and 99% in the second and third parities, respectively. The authors found that the prevalence was higher in heifers than in primiparous cows and that lameness occurred more frequently in Winter and Autumn than in Summer and Spring because the animals were reared under temperate climate conditions, where the cows’ environment may be wetter in Winter and Autumn. The rate of lameness increased by 2%. Regarding the most economically important disorders in dairy cows, mastitis, infertility, and claw and leg disorders are listed [16]. A Tunisian study was conducted on 35 dairy farms to assess the welfare quality of Tunisian Holstein cattle based on several animal welfare indicators validated by the European Welfare Quality Project. The avoidance distance (on the face and in the stall), physical condition, lameness, fertility, somatic cell count (SCC; cells/mL), and milk yield were assessed [17]. The main results showed that the SCC averaged 427.3 ± 90.12 10^3^, being the highest in Autumn and associated with milk yield. Milk yield increased with the number of lactations and varied by the lactation stage. Smaller farms had lower somatic cell counts. The same study reported that the body condition score (BCS) ranged between 1.25 and 4 (lactating cattle) using a BCS scale between one and five. Most cows presented a BC score of 2.5 (50% of cows); however, most dry cows presented a BCS of 2.75 (65% of cows), ranging between a BCS of 1.5 and 4. A BCS of two or less was classified as ‘thin’. The mean number of lactating cows in this category on all farms was 18.9 ± 1.9%. As for lameness, the proportion decreased, with only 19 out of 350 cows (5.4%) showing moderate lameness. Lameness appears to have been the greatest welfare problem within the parameters investigated. In general, the avoidance distances are short, which is an indicator of good human-animal relations and may reflect good farming practices [17]. In dairy herds, some bacterial diseases (*Bovine tuberculosis, Campylobacter enteritis*, Anthrax, Hemorrhagic septicemia, *Mannheimia haemolytica*, and Contagious bovine pleuropneumonia, etc.) are of paramount importance, particularly those considered zoonotic. Among these, *Bovine tuberculosis* (caused by infection with *Mycobacterium bovis*) is perhaps the most problematic. Heritability estimates on the observation and responsibility scales varied between 0.06 and 0.18; the standard errors varied between 0.012 and 0.044. The presumption of inheritance was based on the tuberculin test response and the presence of tuberculosis lesions confirmed during slaughterhouse testing. These results demonstrate that gene truncation can achieve a significant improvement in tuberculosis resistance [18]. *Bovine tuberculosis* (bTB) is considered a major zoonosis in Tunisia and a break to the intensification of production. Tunisia has had a national bTB control program since 1984. It is based on the intradermal tuberculin skin testing of dairy cows and regular meat inspections at the slaughterhouse. Nonetheless, bTB remains prevalent, mainly in the private sector, where disease control is based on sparse veterinary practices and slaughterhouse testing without routine intradermal tuberculin skin testing [19]. In their synthesis on animal health and disease genetics, Berry et al. [20] concluded that the accurate quantification of genetic trends in most health traits is not possible due to the lack of the routine availability of accurate animal health records and data in most countries. Nonetheless, past genetic trends may be predicted based on the estimated genetic correlations with the production traits, and the impact of these correlations can be quantified using the knowledge of past breeding goals.

## 3. Genetic Evaluation of Longevity

Due to its high economic value, longevity is an important part of dairy cattle breeding goals in many countries. Imbayarwo-Chikosi et al. [21] attribute the high economic value of longevity to the herd trait dynamics that depend on the degree of voluntary and involuntary selection. Reducing involuntary culling increases the chances of voluntary culling and keeping high-yielding cows for longer [21]. Longevity employs different trait definitions [22,23]. All of these were based on age at culling, or death (uncensored) or censorship, and survived to the indicated age or period during or between lactations [24,25]. Lifespan appears as a threshold whose expression is not continuous but has a distinct categorical phenotype. Therefore, there are complete and incomplete records for survival dates. Events such as culling, and death are uncensored as they may be known. At the same time, animals may have been lost in pursuit, and no culling or death events are known to have occurred. Animals may also be alive at the time of analysis, so only a lower bound on the productive lifespan is known. Appropriate modeling strategies for such data are required to account for these intrinsic traits without losing the important phenotypic, additive, and environmental dispersal information required for genetic evaluation [26]. According to Vacek et al. [27], different measures for longevity have been reported, such as the length of productive life, total milk production, herd life, the total number of lactations, and survival observed at a certain age, measured from birth or after first calving. Longevity is determined by the voluntary and involuntary culling decisions of individual farmers. In the process of making decisions on culling, the farmers or producers will consider many traits, such as production, health, fertility, and other functional traits such as milking speed, milking temperament, and calving ease [28]. Longevity has been excluded from breeding programs because the genetic evaluation of this trait is generally difficult. In several studies, researchers have attempted to address this issue using various models that have been proposed to assess cattle longevity [29,30]. Gene scoring systems are not standardized by country because lifespan can be measured in many ways. Many models have been developed for the genetic assessment of lifespan. Ducrocq et al. [30] have been adopted by many researchers worldwide to analyze the survival characteristics of dairy cows. This model attempts to estimate the probability that an animal will survive to time “*t*”, given that it survived to time “*t* − 1”. A second genetic assessment model for survival is the multi-trait animal model (MT). Here, a trait is defined as survival (0 or 1) to a specified endpoint or within a specified time interval during the cattle’s lifetime [31]. Veerkamp et al. [32] proposed a longitudinal generalization of multiple-traits models for survival achieved through a random regression (RR) model. Binary observations (0 = culled, 1 = survived) are assigned to each discrete unit in the cow’s lifetime, such as per lactation or month after first calving. Furthermore, survival is generally considered a genetically different trait in different parities, even in different stages of lactation [33]. Linear animal and sire models have been used by several researchers [34,35]. In practice, the performance of a model depends on the definition of the survival trait and the quality of the data [36], and the strategy of processing the merit index, combining single-trait models or multivariate models for groups of traits [37]. In Tunisia, the lifespan of Holstein cows was expressed by the number of lactations initiated in dairy cows [8]. M’Hamdi et al. [5] showed that the lifespan of Tunisian dairy cows is considered persistent, defined as the probability that a cow will live to that age if given the chance to live to that age. He used the Weibull proportional hazards model. The Weibull distribution results from a combination of the simplicity of the Weibull survival function SUR_0_ (*t*) = exp (−(ht) ρ) and its flexibility. Weibull regression can model constant (ρ = 1), increasing (ρN1), and decreasing (ρb1) hazards. Further analysis is greatly facilitated if *h*_0_(*t*) can be approximated. 

The following Weibull model was used:$$h\left(t,z\right)=h_{0}\left(t\right)*e^{\left(hys+plst+ml+yb+age1+sire\right)}$$
where:

*h(t, z)* = hazard function of the cow at time *t*; *h_0_*(*t*) = Weibull baseline hazard function at time *t*; *hys* = random time-dependent effect of the herd–year–season; *plst* = fixed time-dependent effect of the parity–stage of lactation; *ml* = fixed time-dependent effect of the class of milk production expressed as deviations within the herd–year; *yb* = fixed time-dependent effect of annual change in herd size; a*ge*1 = time-independent effect of age at the first calving; and *sire* = random time-independent effect of the sire of the cow.

The average length of productive life was 37.5 months, and the heritability was 0.19. Ben Salem et al. [38] calculated the longevity or true herd as the number of days from the first calving to the culling or censoring; they found an average value of 48.6 ± 28.2 months. Ajili et al. [8] analyzed the true herd life (THL) variations for the last lactation of a cow. Longevity corrected for voluntary culling is called functional longevity, whereas observed longevity is called true longevity or true herd life; they found an average THL of 41.9 months.

The linear model used to study the herd life was:$$THL_{ijklmn}=\mu +HY_{i}+S_{K}+NL_{i}+age_{m}+e_{ijklmn}$$
where:

*THL_ijklmn_* is the true herd life µ: is an overall mean, *HYj* is the fixed effect of the herd by calving year *j, Sk* is the fixed effect of calving season k, *NL* is the fixed effect of lactation number, *Age_m_* is the fixed effect of age at first calving, and e_ijklmn_ is a residual effect.

The longevity from the Tunisian studies varies between 37.5 [5] and 48.6 months [38]. These results agree with those observed by Morek-Kopeć et al. [39] in Polish Simmental Cattle (39.3 months) and are greater than the results of Kern et al. [40] in Brazilian Holstein cows (33.5 months). The heritability is low (0.12–0.19) and in the range of the literature worldwide, as reviewed by Imbayarwo-Chikosi et al. [36], as well as the recent results reported by Ghaderi- Zefrehei et al. [41] in Iran (0.086); Wiebelitz et al. [42], (0.05) and Van Pelt and Veerkamp [43], (0.01) in Germany; and Kern et al. [44] in Brazil (0.15).

## 4. Genetic Evaluation of Fertility

Female fertility is one of the most important factors affecting the longevity of dairy cows. Many authors have reported antagonistic genetic relationships between reproductive and milk production traits [45,46,47,48]. There is abundant evidence linking the selection for milk production and infertility [49]. Higher milk yields are genetically correlated with longer calving intervals, longer days to first service, and lower conception rates at first service. An evaluation of the reproductive parameters in Tunisian Holsteins was conducted on 35 dairy farms [50]. The main fertility characteristics used were calving interval (CI), days open (DO), days to first visit (DFS), and visits per conception (NSC). The fertility characteristics averaged 444 ± 101.5, 154 ± 78.4, 82 ± 56.8 days, and 2.1 ± 1.0 for CI, DO, DFS, and NSC, respectively. The average age of the cow was 6.0 ± 1.0 years of age. The heritability estimates were low: 0.03, 0.06, and 0.09 for DFS, DO, and CI, respectively. A Bayesian analysis yielded similar results [51]. The posterior mean heritabilities for CI, DO, DFS, first service to conception interval (FSC), and NSC were 0.047, 0.03, 0.025, 0.024, and 0.069, respectively. The reproducibility of the same respective functions was 0.106, 0.094, 0.051, 0.036, and 0.17 (Table 1).

The estimated genetic correlation between the calving intervals and DO is the highest (0.85). The lowest estimated genetic correlation (0.25) was found between DFS and NCS. Makgahlela et al. [54] reported 0.074, 0.076, 0.044, 0.058, and 0.046 for CI, DO, FSC, DFS, and NCS, respectively. The heritability estimates for the reproductive traits in the current analysis are comparable to those of M’Hamdi et al. [11]; using a multi-trait animal model, these authors reported estimates of DFS, DO, and CI of 0.032, 0.041, and 0.063, respectively. However, the results of the current heritability estimation study are less than those of Sewalem et al. [28]; performing a bivariate analysis, we found heritability estimates of 0.08 and 0.05 for DFS and FSC, respectively. Yamazaki et al. [48] used a multi-trait linear model to analyze the reproductive traits of Japanese Holstein cattle, and the heritability values for DO were 0.07 in the first lactation, 0.06 in the second lactation, and 0.12 in the third lactation. Agili et al. [8] found consecutive deliveries and intervals from delivery to first service, with release days of 427, 90, and 163 days, respectively. The data included his 128.652 records collected between 1990 and 2004 on his 47,276 Holstein Friesian cattle in 142 herds. The same authors reported positive phenotypic correlations between the fertility traits and true herd life (Table 2).

The estimates of the genetic parameters for female fertility traits in Tunisian dairy herds are low, even lower than those reported in the literature. In short, to improve the reproductive performance of Tunisian Holstein breeds, selection should not only focus on fertility traits, but also on improving cattle reproductive management. The genetic correlations between health and fertility are good, and the selection for mastitis or lameness is associated with shorter calving intervals, shorter intervals from calving to first service, fewer inseminations to conception, and higher rates of non-reversion. This suggests that it is likely to lead to an increase [1]. Poor health conditions such as mastitis and lameness have been shown to adversely affect fertility [55]. Early lactation illness is thought to affect the cow’s ability to show fever (thereby reducing detection) and may become pregnant after insemination. Huszenicza et al. [55] found that mastitis can impair the resumption of ovarian activity in postpartum cows. B. Mastitis cows had delayed luteal activity and decreased estrous behavior after 15–28 days of milking.

## 5. Association of Longevity with Fertility and Type Traits

### 5.1. Fertility

Reproductive disorders reduce fertility, prevent conception, cause problems in delivering healthy calves, cause postpartum complications, increase inter-calving intervals, reduce milk yield, and affect overall life expectancy [56]. Among the many factors that influence the age of replacement in primiparous cows, reproductive disorders are particularly important [57]. Under Tunisian conditions, the optimal primal age was 23–27 months. Reducing the age of replacement to about 24 months may improve the 305-d and longevity yields and extend the swarm longevity [58]. Medium-performing cows tended to stay in the herd longer than low-performing or high-performing cows. Shorter-than-expected residence times in prolific cows can be explained by selection for reasons other than production (involuntary selection), such as poor health and reduced fertility. These results are comparable to those by Ducrocq [30] and Weigel et al. [26]. The phenotypic correlations between the actual herd life and the milk, fat, and protein yields ranged between −0.04 (open day) and 0.06 (calving interval) on the fertility parameters [8]. Environmental factors (year of calving, season of calving) and management factors (herd) are very important sources of variability in milk production, reproductive traits, and thus herd lifespan. A combination of clearly defined reproductive goals and better management (such as selection for low production, selection of non-yield traits, and diversified feed sources) improves the performance of Tunisian Holstein cattle [8]. Functional lifespan (FL) or productive lifespan (LPL) length is an important trait for measuring the overall functional fitness in cattle. The effect of age at first calf on the milk yield and actual herd life span was investigated in Tunisian Holstein cattle. Ajili et al. [8] investigated 33.407 first lactating records for cows born between 1987 and 2001 from 166 herds and found that the herd life expectancy was 38.6 months (SD = 24 months) and the mean age at first birth was 28.7 months (SD = 3.4 months). The backward mean heritability for age at first delivery was 0.08.

### 5.2. Type Traits

Type traits are currently being measured as part of the genetic improvement programs aimed at linking trait types with milk production, conformation, reproduction, and longevity. The purpose of including type traits is to improve the conformation of cattle by providing better body and functional and reproductive structures so that they can meet the challenges of increased production [3]. Linear traits are used to select for longevity, mainly because all trait types are acquired early in cattle life and have moderate heritability [59,60]. Selecting the types of traits associated with herd longevity can be beneficial in reducing involuntary selection and increasing profitability [61]. Type traits that have a significant impact on a cow’s longevity are those associated with the udder, feet, legs, and leg sections, for example, anterior attachment, texture, depth, posterior mammary attachment height, posterior mammary attachment width, median ligament, and bone quality. and trunk angle [28]. The selection of the hind mammary width and height, mammary structure, mammary cleft, loin strength, bone quality, and final score may improve the longevity and milk production [40]. Zavadilová and Štípková [35] found a positive genetic correlation between longevity and type traits, BCS (0.14–0.19), tail angle (0.15–0.21), and hook quality (0.05–0.19). A slightly weaker correlation was found in the results (−0.13–0.02). Both the true and functional longevity were tested, and the type traits showed stronger genetic correlations with functional longevity. Kern et al. [44] found similar results for the type traits. The correlation between lifespan and paw angle was in the same range of −0.18 to 0.08 [44] and −0.10 to 0.10 [35]. The correlation between the type characteristic and udder depth was positive, with the longevity characteristic 0.20–0.27 (that is, the taller the breast, the longer the longevity) [62], 0.04–0.11 [35], and 0.17–0.31 [45]. Zavadilová and Štípková [35] found a negative correlation with the median ligament type trait (−0.19). [61] found a positive correlation (0.28) for the breast-supporting trait, which is considered to be the same as the median ligament, and a positive correlation (0.17–0.29) for the posterior papillary trait. Setati et al. [62] observed a low heritability of longevity and moderate heritability of most types of traits, except for lump height and pre-papilla length. All of the phenotypic correlations between lifespan and linear traits were slightly positive (0.01–0.09). The genetic correlations between longevity and breast features and angles were moderate-to-high and positive (0.22 to 0.48). In conclusion, the positive genetic correlations and moderate heritability suggest that the selection of udder features and angles could improve longevity in dairy cows [63,64].

## 6. Genetic Evaluation Enhanced with Genomic

During the last century, dairy cattle breeding schemes were based on the phenotypic data processed according to quantitative genetic rules. Evaluation accuracy requires heavy and expensive phenotype recording and the progeny testing of bulls by observing the performances of their daughters. Indeed, the application of quantitative genetics to dairy cattle breeding between 1960 and 2007 was very successful, increasing the milk yield and profitability of the production systems [65]. In Holstein dairy cattle, milk production still increases by 110 kg per animal per year [66]. This process involved thousands of cows and took many years to complete. However, the use of molecular information to make selection decisions in breeding schemes was envisaged decades ago. In the 1990s, DNA analysis marked the beginning of the new field of genomics [67].

First, a genetic marker approach based on the concept of the detection linkage relationships between genomic regions and quantitative traits-quantitative trait loci (QTLs)was utilized in the marker-assisted selection (MAS) process. The usual MAS idea is a three-step process: (i) detect one or more QTLs; (ii) find the causative gene (causative mutation); and (iii) increase the frequency of favorable alleles by selection or introgression. Except for a few cases (halothane in pigs, Barolo in sheep), the impact of MAS in livestock breeding programs was minimal (10%), as a QTL above the selected significance level can usually explain only a fraction of the trait variance [67]. In the context of multi-trait breeding goals, such markers may have a less overall impact on the breeding goals, as stronger responses to one trait often appear at the expense of another trait. The MAS approach was successful for traits with simple genetic determinism but had disappointing results in more complex situations [68].

Second, the recent availability and affordability of high-density panels of single nucleotide polymorphism (SNP) markers have opened up this new possibility. Mészáro et al. [69] proposed what is now known as genomic selection. In general, 45,000 SNPs are used in bulls and 3000 SNPs are used in genotyping cattle, heifers, and calves. Genomic selection has revolutionized dairy farming in the last decade and is considered a success story [68]. High-density SNP genotyping is used in two ways: Genome-Wide Associations studies (GWSA) to identify genetic markers (SNPs) or genomic regions (QTLs) associated with traits. This technology has been largely used to detect the genetic associations with farm animal traits, mainly with low heritability, to achieve faster genetic progress. Hundreds of markers were identified to be associated with production, functional, and novel traits in dairy cattle. Implementing a GWA study on the Holstein breed, Nayeri et al. [70] identified interesting markers associated with longevity (*SYT10* on chromosome 5, *ADAMTS3* on chromosome 6, *NTRK2* on chromosome 8, and *DERL1* and *SNTG1* on chromosome 14). Tiezzi et al. [71] concluded that the most significant SNP associated with longevity traits were found on Bos taurus autosome 18 (*BTA18*) and were mostly located within the QTL regions associated with mastitis. In contrast, for the lifetime profit index, the strongest associations were detected on *BTA14* and *BTA18*. They also observed an association between *BTA6* and *BTA20* for lactation persistency. For clinical mastitis, which is related to longevity, Bermotiene [72] observed in Holstein’s first lactation an association between genetic variation and the regions on chromosomes 2, 14, 20, and 29. In Tunisia, no QTL mapping experiment has ever been conducted on dairy cattle. Recently, a study aimed to analyze the polymorphism of the gene responsible for the biosynthesis of leptin using the polymerase chain reaction-restriction fragment length polymorphism (PCR-RFLP) technique. Indeed, leptin is a glycoprotein that is involved in the defense mechanisms of the mammary gland of dairy cows. The level of this protein secretion rises significantly in response to viral or bacterial infections [73]. A total of 160 blood samples were collected from dairy Holstein breed cows. The results showed the presence of two alleles A and B and three genotypes AA, AB, and BB, with a dominance of the allele A. The results indicate that animals carrying the BB gene could contribute to a reduction in somatic cells in cattle. Based on this observation, they are more resistant to mastitis. This peptide can be considered a candidate gene for udder health [74].

The whole-Genome selection presents numerous advantages versus the traditional quantitative approach: (i) genomic information is not affected by environment; (ii) it can be available at an early age; (iii) it can be obtained on all selected candidates; (iv) it enhances the reliability in predicting the mature phenotype; and (v) the genomic approach helps us to select for a wide range of traits and, in turn, saves time and effort [75]. The ‘development of inexpensive high-throughput genotyping platforms of SNP markers revolutionized dairy cattle breeding” [76]. In the last decade, genomic selection has been very successful and was quickly adopted in the largest populations worldwide; genetic trends have doubled, especially for traits with low heritability. Several countries have already adopted genomic selection in the dairy cattle industry and an international genomic evaluation model across countries was implemented (Genomic MACE). Nine years after the adoption of genomic selection in the United States of America (US), the results were very successful. A dramatic reduction in generation interval (GI) was observed. The GI of the Sire of offspring decreased from seven years to less than 2.5 years, and the GI of the fathers of daughters decreased from about four years to almost 2.5 years. The difference is relatively stable. The most dramatic responses to genomic selection were observed for the inherited traits of low daughter pregnancy rate (DPR), productive life (PL), and SCS. Genetic propensities have gone from near-zero to large and favorable, resulting in rapid genetic improvements in fertility, longevity, and health; however, these traits have eroded over time. These results demonstrate a positive effect of genomic selection in US dairy cows, although this technique has only recently been used. Based on a four-way selection model, the genetic gains per year ranged between approximately 50–100% for yield-related traits and 3- to 4-fold increases for low-heritability traits [76]. Similar results were observed in other countries and dairy cattle populations. The unique genomic research program initiated on cattle in Tunisia was implemented on the phylogenic insights into the history of Tunisian native cattle. Furthermore, Ben Jemaa et al. [76], in a genome-wide characterization of local cattle breeds from the central and western Mediterranean, pinpointed the admixed origin of the genome of the Tunisian native cattle population with the two main European and African ancestries. However, Ben Jemaa [76] suggested the implementation of a genomic selection program for Tunisian dairy cattle.

## 7. Conclusions

The main conclusions of this review are:

The genetic parameters observed for the Tunisian Holstein dairy cattle are in agreement with the literature used in the present manuscript.

During the last few decades of the twentieth century, the application of quantitative genetics to dairy cattle breeding was very successful, increasing the milk yield and profitability of the production systems.

At present, herd management focuses on the balance of functional traits (particularly health, fertility, longevity, and welfare) against production.

The performance of an evaluation model depends on the definition of the trait, the quality of the data, and the strategy of processing the merit index, combining single-trait models or multivariate models for groups of traits.

However, the revolution in genotyping provided by high-density SNP coupled with a cost reduction has resulted in large databases of individuals with genome-wide genotypic data. Moreover, in the last decade, genomic selection has been very successful and was quickly adopted in the largest populations worldwide; genetic trends have doubled, especially for traits with low heritability.

In Tunisia, Genomic research is sporadic, with no implementation of a genomic selection program to date. Despite the success of several countries in increasing the annual genetic gain of different traits of economic interest, we recommend to Tunisia, as it is a member of the ICAR consortia, to reinforce the phenotypic recording system and extend it to functional parameters, mainly longevity, and initiate a genomic selection program. The implementation and validation of the genetic evaluation process, in the INTERBULL framework, would enable Tunisia and developing countries in a similar situation to overcome the gap caused by genomic selection.

## Figures and Tables

**Table 1 genes-14-00670-t001:** Main results on genetic parameters of milk yield, reproduction, and longevity in Tunisian Holstein dairy cows.

Traits	Genetic Parameters				Model	Reference
	Heritability			Repeatability		
Test-Day (Milk)	1st lact	2nd Lact 3rd lact	All lact.		Test Day Animal Model/Bayesian	[51]
5	0.129	0.132	0.141			
35	0.087	0.127	0.128			
65	0.058	0.118	0.115			
95	0.064	0.112	0.105			
125	0.093	0.110	0.100			
155	0.125	0.111	0.095			
185	0.154	0.116	0.098			
215	0.174	0.128	0.012			
245	0.180	0.148	0.174			
275	0.166	0.166	0.244			
305	0.146	0.172	0.287			
Milk yield-305 d	0.14	0.12	0.13		Multiple-trait AM/Bayesian	
Fat yield-305 d	0.17	0.15	0.16			
Prot yield-305 d	0.20	0.17	0.18			
Milk yield-305 d	0.273 (0.020)				Multiple-trait AM/Bayesian	[52]
Fat yield-305 d	0.198 (0.010)					
Prot yield-305 d	0.187 (0.010)					
Milk yield-305 d	0.210 (0.050)				Multiple-trait AM/REML	[52]
Fat yield-305 d	0.159 (0.040)					
Prot yield-305 d	0.158 (0.040)					
Milk yield-305 d	0.230 (0.020)			0.38 (0.01)		[53]
CI	0.047 (0.013)			0.106 (0.026)	Multiple-trait AM/Bayesian	[52]
DO	0.030 (0.010)			0.094 (0.023)		
DFS	0.025 (0.009)			0.051 (0.013)		
FSC	0.024 (0.007)			0.036 (0.009)		
NSC	0.069 (0.010)			0.170 (0.026)		
CI	0.063			0.152	Multiple-trait AM/REML	[5]
DO	0.041			0.135		
DFS	0.032			0.128		
NSC	0.027			0.034		
DO	0.090				Single trait AM/REML	[50]
DFS	0.060					
Longevity	h^2^ Log scale	h^2^ Original scale	Weibull distribution		Proportional hazard (PH) model	[17]
	0.123	0.190	1.21			

CI: Calving Interval; DO: Days Open; DFS: Days to First Service; FSC: First Service to Conception. NSC: Number of services per conception; (…): SD: standard deviation.

**Table 2 genes-14-00670-t002:** Phenotypic correlation between true herd life and milk and reproductive traits of Holstein Frisian cows in Tunisia.

	Milk Traits (305 d)			Reproductive Traits		
	Milk	Fat	Prot.	CSI	DO	CI
**THL**	0.07 **	0.11 **	0.09 **	0.04 **	−0.03 *	0.06 **

THL: true herd life; CI: calving interval; CSI: calving to first service interval; DO: days open; * *p* < 0.1; ** *p* < 0.01 (Ajili et al., 2007 [8]).

## Data Availability

Data was used in the study.

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
