# Peer review of "Assessment of Genetic and Health Management of Tunisian Holstein Dairy Herds with a Focus on Longevity"

_genes, 2023, doi:10.3390/genes14030670_

Round 1

Reviewer 1 Report

The health and welfare of all animals in the dairy herd are paramount for profitable and efficient milk production. For the modern dairy cow, advances in genetics and breeding for productivity have resulted in an increased incidence of health disorders and reduced longevity. However, to maintain farm sustainability, farmers need to optimize the balance between maximum production and minimum production costs. This paper studied the influence of genetic factors on some diseases of Tunisian Holstein cattle, reviewed the main achievements of genetics and health management of Tunisian Holstein cattle, and reviewed the relevant situation at home and abroad. Results showed that the risk of culling increased with common diseases. Herd management focuses on the balance of functional traits (Especially Health, fertility, longevity, and welfare) against production. The authors conclude with a recommendation that Tunisia strengthen the phenotypic recording system and extend it to functionality, primarily longevity, and initiate a genome selection program. The manuscript was well written, the objectives were clear, the analyses were done comprehensively, and the results and the discussion were detailed. Apart from this, with some minor revisions, the manuscript can be accepted for publication. The detail of my considerations is presented below.

Minor concerns:

Point 1: Line 91: The format in which numbers are written in the text should be consistent throughout the text.

Point 2: Line 108: Percentages should be written in a uniform format throughout the text.

Point 3: Line 264: Write "P" in italics.

Point 4: Line 370-374: Genes are written in italics. Please keep the text consistent.

Point 5: Line 560-562: Please keep the text consistent.

Author Response

Dear reviewer

Thank you for the add velue comments

Reviewer 2 Report

LIne 41 – 42 Brickell & Wathes [4] suggested that extending the productive life of cows reduces 41 replacement costs, allows for more limited selection, and yields more potential milk yields 42 from adult cows, thus improving milk production. It claims to be more eficiente. This sentence must be justified

Line 50 - In addition, Bennett [7] reports that mastitis and lameness are the most costly 50 diseases for dairy farmers. This sentence must be justified

Line 75   -   cost-effective alternative – how the cost?

Line 128 - some bacterial diseases – how diseases?

Line 168 - certain age – how age?

Line 281 - Reproductive failure causes significant economic losses by increasing age at first 281 calving (AFC) and calving interval (CI), increasing the cost of insemination and Exchange – how losses?

Line 354minimal – how minimal?

Author Response

Dear reviewer

Thank you for your comments 

They are very important to improve our paper

BR

Reviewer 3 Report

Great review article. Please, review my comments on the attached file. 

Best regards.

Author Response

Dear reviewer 

Thnak you for your valuable comments

All coorections are made in the text and in the cover letter for reviewers

Cordially
